# Pitahaya Peel: A By-Product with Great Phytochemical Potential, Biological Activity, and Functional Application

**DOI:** 10.3390/molecules27165339

**Published:** 2022-08-22

**Authors:** Sandra N. Jimenez-Garcia, Lina Garcia-Mier, Xóchitl S. Ramirez-Gomez, Humberto Aguirre-Becerra, Alexandro Escobar-Ortiz, Luis M. Contreras-Medina, Juan F. Garcia-Trejo, Ana A. Feregrino-Perez

**Affiliations:** 1División de Ciencias de la Salud e Ingeniería, Campus Celaya-Salvatierra, C.A. Enfermedades no Transmisibles, Universidad de Guanajuato, Av. Ing. Javier Barros Sierra No. 201 Esq. Baja California, Ejido de Santa Maria del Refugio Celaya, Guanajuato 38140, Mexico; 2Departamento de Ciencias de la Salud, Universidad del Valle de México, Campus Querétaro, Blvd, Juriquilla No. 1000 A, Delegación Santa Rosa Jáuregui, Santiago de Querétaro 76230, Mexico; 3División de Estudios de Posgrado, C.A. Bioingeniería Básica y Aplicada, Facultad de Ingeniería, Universidad Autónoma de Querétaro, C.U. Cerro de las Campanas S/N, Colonia Las Campanas, Santiago de Querétaro 76010, Mexico; 4Facultad de Química, Universidad Autónoma de Querétaro, C.U. Cerro de las Campanas S/N, Colonia Las Campanas, Santiago de Querétaro 76010, Mexico

**Keywords:** phenolic compounds, biological activity, *Hylocerous* spp.

## Abstract

*Hylocereus* spp. present two varieties of commercial interest due to their color, organoleptic characteristics, and nutritional contribution, such as *Hylocerous polyrhizus* and *Selenicerus undatus*. The fruit recognized as dragon fruit or Pitahaya is an exotic fruit whose pulp is consumed, while the peel is discarded during the process. Studies indicate that the pulp has vitamin C and betalains, and seeds are rich in essential fatty acids, compounds that can contribute to the prevention of chronic non-communicable diseases (cancer, hypertension, and diabetes). In the present study, polyphenolic compounds, biological activity, and fatty acids present in the peel of the two varieties of pitahaya peel were evaluated, showing as a result that the variety *S. undatus* had higher antioxidant activity with 51% related to the presence of flavonoids 357 mgRE/g sample and fatty acids (hexadecanoic acid and linoleate) with 0.310 and 0.248 mg AG/g sample, respectively. On the other hand, *H. polyrhizuun* showed a significant difference in the inhibitory activity of amylase and glucosidase enzymes with 68% and 67%, respectively. We conclude that pitahaya peel has potential health effects and demonstrate that methylated fatty acids could be precursors to betalain formation, as well as showing effects against senescence and as a biological control against insects; in the same way, the peel can be reused as a by-product for the extraction of important enzymes in the pharmaceutical and food industry.

## 1. Introduction

The pitahaya is an exotic fruit belonging to the Cactaceae family and the *Hylocerous* genus [1,2]. Originally from Latin America and western India, it has two highly commercial varieties (*H. polyrhizus* and *undatus*), the most cultivated being *Selenicerus undatus*, which was previously classified as *Hylocereus* [3]. In Mexico, it is grown as in many other regions of the world such as Israel, Nicaragua, and Vietnam. In Mexico, they are grown mainly in Yucatan, Puebla, and Guanajuato. A total of 95% of the world’s production focuses on the *S. undatus* variety. The pitahaya fruit, or dragon fruit as it is also known, is made up of the edible part that includes seeds and pulp of different colors that range through shades of yellow, orange, red, or purple. Numerous benefits have been attributed to it, in addition to its being very attractive due to its nutritional value, vitamin C, and antioxidants (betalains, polyphenols among others), as well as a certain amount of oils that are always present in more significant amounts in the seeds of the fruit [4]: the reason why it is widely used in the preparation of jellies, juices, jams, and marmalade, in addition to its fresh and dehydrated composition. The inedible part, which is the peels that cover the pulp, represents 33% of the total weight of the fruit and is considered an unusable residue; however, various studies indicate that it can be a good source of pigments to use in various industrial processes [5,6].

Using residues derived from industrial processes, mainly those of fruits and vegetables, given their content of bioactive compounds, it is increasingly attractive to obtain by-products with added value from them [7]. Obtaining bioactive compounds from waste represents economic value. Furthermore, it is important since it is related to potential health benefits [8]. Antioxidants especially secondary metabolites, due to their abundance in fruits and vegetables, are the compounds that are found in a more significant proportion in the peels of fruit or vegetables since their function is to protect them from solar radiation and pests or insects, as well as protect them from diseases [9]. Furthermore, the presence of these compounds in the pulps is important since they assimilate the nutrients of the plants, and protein synthesis, among other functions; and when consuming the fruit, the secondary metabolites such as phenolic compounds, several studies have reported medicinal characteristics related to the prevention of chronic non-communicable diseases such as cancer, diabetes, hypertension, cardiovascular, and gastrointestinal [8]. Other investigations showed that the main mechanism of these activities is carried out through the transfer of electrons, reducing oxidative stress. Therefore, antioxidant activity is one of the most reported properties of the pitahaya fruit; however, it varies depending on the variety, origin, and degree of ripening [10,11,12,13]. The antioxidant power of pitahaya is attributed to the presence of betalains in addition to phenolic compounds [10].

Phenolic compounds are mostly secondary metabolites in the plant metabolism, for example, they are composed mainly of simple structured compounds (phenolic acids to more complex polyphenols such as flavonoids) [8]. Phenolic compounds are identified for their nutraceutical properties and benefits on chronic non-communicable diseases, such as cancer prevention, antioxidant activity, and antibacterial, antihypertensive, and antidiabetic properties, among other beneficial effects. Hence, they are considered an important source to be applied medically or pharmaceutically [14,15]. In addition, the peels and seeds of the pitahaya fruit have been used to prepare foods such as juices, jellies, ice cream, etc. Pitahaya has high amounts of carbohydrates, sugars, and some micronutrients such as zinc and iron; it has been shown that red pitahaya increases hemoglobin levels when consuming red pitahaya juice [16]. On the other hand, it has been proven that essential fatty acids have been found in the pulp that help reduce cholesterol and lipoproteins. Consequently, foods that contain functional properties and high nutritional content are considered an alternative for the prevention of chronic non-communicable diseases such as arterial hypertension and diabetes mellitus, which are two diseases that are directly related and require adequate and careful supervision. Hypertension not only causes heart attack, heart failure, and stroke, but in many cases, it also often causes diabetes mellitus [17]. Thus, type II diabetes mellitus has converted into a topic of interest for scientific groups related to these topics. In most of the diabetic population, 80% of diabetic patients have complications with hypertension, and vice versa, 70% of hypertensive patients may experience impaired glucose tolerance or develop type II diabetes mellitus. The prevalence of diabetes is increasing around the world; this disease is one of the main public health problems [18]. Type II diabetes mellitus is distinguished by showing resistance to insulin and the abnormalities present in the β cells, presenting a decrease in the production and secretion of insulin. Dietary and antioxidant components of foods show an important role in glucose homeostasis. Consequently, the functionality of the use of bioactive compounds (phenols and flavonoids) shows a decrease in blood glucose levels related to the inhibition of enzymes in the carbohydrate metabolism (α-amylase and α-glucosidase). As a result, inhibition of α-amylase and α-glucosidase is a key therapeutic objective in the management of type II diabetes mellitus. Furthermore, oxidative stress causes the formation of free radicals that trigger insulin resistance, an adverse effect present in diabetes [17]. In the past f few years, studies focused on vegetables with functional compounds with a biological effect on health have increased, not only by looking at the pulp but also by reusing the fruit peel and thus being able to integrate it into an alternative to the modern medical system [19]. Hence, the present investigation is focused on comparing the bioactive parameters in pitahaya peel (*H. polyrhizus* and *S. undatus*) to be used as a by-product in the pharmaceutical, medical, or food industry. The results will offer information on the bioavailability of functional compounds, on the stability and resistance of enzymes that have properties in chronic non-communicable diseases such as type II diabetes mellitus, antioxidant properties, and beneficial effects on the angiotensin-converting enzyme and cardiovascular problems, as well as on how to provide greater knowledge about the profile of metabolites present in the pitahaya peel of different species.

## 2. Results and Discussion

### 2.1. Quantification of Phenolic Compounds in Pitahaya Peels

The profile of total polyphenolic compounds provides an approximation of bioactive compounds present in plants, which can have a protective effect on chronic diseases such as diabetes and hypertension, among others. Concerning the total phenolic compounds, the analysis of variance indicates that there is a significant difference between the peels of the two varieties evaluated. The concentration of *H. polyrhizus* with 1.863 mg GAE/g sample in a peel is lower than *S. undatus* with 2.717 mg GAE/g sample.

On the other hand, the concentration of total flavonoids is slightly higher in *S. undatus* compared to *H. polyrhizus*, with 356. 74 and 352.09 mg RE/g sample, respectively, which does not show a significant difference according to the analysis, as indicated in Table 1. The trend is maintained in the data obtained for tannins, where *S. undatus* (2.886 mg RE/g sample) has a higher concentration compared to *H. polyrhizus* (1.628 mg CE/g sample) showing a significant difference according to the statistical analysis (Table 1). The results indicate the various influences on the concentration of secondary metabolites with biological activity.

Foods of plant origin are sources of compounds with biological activity; however, a large part of this raw material is wasted as it is not consumed or eliminated during various transformation processes. It is estimated that approximately 1.3 billion tons of food, food by-products, and waste, including agro-industrial waste such as fruit peels, are wasted per year [7,20]. In the last half-decade, the use of industrial by-products such as bagasse and fruit pomace to produce reduced-calorie snacks has exploded their use, while the use of fruit peels has been more limited, focusing mainly on the obtention of pectin [21,22]. The peels of the fruits represent a significant percentage of the total mass of the product, and in the case of the pitahaya peel, they represent approximately 33% of the total weight of the fruit, in addition to being an integral part of it, which makes them a source potential of nutrients and compounds with biological activity that can be used in various sectors such as pharmaceuticals, cosmetics, and food. The pitahaya peel (S. undatus), due to its anthocyanin content, mainly betacyanin, is considered an ideal candidate for obtaining dyes that can be used in the food and cosmetology industries [23]. However, anthocyanins are not the only compounds present in the pitahaya peel; the presence of polyphenolic compounds is associated with various biological activities, among which the antioxidant property stands out, which can vary depending on the type of polyphenol present in the peel. The results indicate that the peel of S. undatus has a significantly higher amounts of total polyphenols compared to the peels of H. polyrhizus (389.017 and 371.003 mg /g samples, respectively). The phenolic compounds present in both the peels of S. undatus (2.717 mg GAE/g sample) and H. polyrhizus (1.863 mg GAE/g sample) are higher than those reported by Nurliyana, Syed Zahir, Mustapha Suleiman, Aisyah, and Kamarul Rahim [16] for an ethanolic extract of H. undatus peel (36.12 mg GAE/100 g), as well as what was reported by Som et al. [24] for methanolic and chloroform extracts of H. undatus peel (48.15 and 18.89 mg GAE/100 g extract, respectively), indicating that the affinity of the solvent with the compounds present in the extracts influences the extraction of the compounds of interest. Similarly, studies indicate that the state of maturity of the fruit influences the content of nutrients present in Hylocereus sp. [25,26]. The peel of various fruits is also a source of tannins, which represent an important group of phenolic compounds. They can form complexes with proteins, nucleic acids, and polysaccharides, among others. Tannins can be classified into hydrolyzed and condensed, the latter being the precursors of anthocyanidins [27]. The pitahaya peel shows a significant difference between S. undatus 2.886 mg CE/g sample and H. polyrhizus 1.628 mg CE/g sample, data that are higher than those reported by Suleria et al. [28] where a concentration of condensed tannins of 0.03 ± 0.01 mg CE/g sample for a variety of Australian pitahaya is observed. The difference can be attributed to the conditions of the cultivation of the pitahaya, extraction technique, and storage. The content of condensed tannins present in the pitahaya peel is like that reported for other fruit peels such as banana peel (1.22 mg CE/g sample), apple peel (2.25 mg CE/g sample), pineapple peel (1.23 mg CE /g sample), products that present an acceptable degree of astringency to the palate [28].

On the other hand, both the peels of *S. undatus* and *H. polyrhizus* samples present a higher concentration of flavonoid-type polyphenols (356.74 and 352.09 mg RE/g sample, respectively) compared to phenols and tannins. Studies indicate that flavonoids show a great capacity to capture free radicals, which contributes to the efficiency of the enzymatic and non-enzymatic antioxidant system, suppressing the formation of free radicals such as superoxide ion, hydrogen peroxide, hydroxyl radical, among others, and thereby counteracting oxidative stress [29,30]. Oxidative stress is the result of a series of chain reactions that contribute to the development of various diseases, including diabetes mellitus and hypertension, among others.

### 2.2. Identification of Gas-Chromatography-Mass Spectrometry (GC-MS) Analysis

Fats play an important role in food. The fatty acid composition is a complex mixture of saturated, monounsaturated, and polyunsaturated compounds with various carbons and different chain lengths. The analysis of the profile of methylated fatty acids (FAMEs) showed the presence of three methylated fatty acids that could be quantified with the technique used and corroborated with the library of reference; the other fatty acids identified in the chromatogram had a percentage of similarity to the standard ≤ 10%; therefore, only three fatty acids identified in the library could be quantified, which were quantified as shown in Figure 1 and Figure 2. The quantification of the identified fatty acids indicates a difference between peels *H. polyrhizus* and *undatus* in the content of the total methylated fatty acids shown in the Table 2. Only in the methyl ester of undecanoic acid is no significant difference observed between the two varieties. On the other hand, hexadecanoic acid methyl ester is significantly different in *H. undatus* with 0.310 mg AG/g of the dry sample compared to *H. polyrhizus* with 0.104 mg AG/g of dry sample; in the same way, in acid methyl linoleate, with a 0.248 mg AG/g, 0.087 mg AG/g of dry sample in *H. undatus* and *polyrhizus*, respectively. On the other hand, other compounds such as alkanes, alkenes, and amides were identified, and these could not be verified with the standards since a FAMEs standard was used and they were only verified with the library of the compounds found, the hexadecanamide related to a fatty amide, which is the carboxamide derived from palmitic acid where we could assume that it is being synthesized to form a fatty acid.

Other authors show that three fatty acids in the pitahaya peel that were identified are Hexadecanoic acid or palmitic acid > methyl linoleate > undecanoic acid, all of them involved in the metabolism of the plant in addition to contributing to the attraction of pollinating insects, as well as repelling insects and pests. The *S. undatus* variety presents a higher proportion of the identified fatty acids compared to *H. polyrhizus*, except for undecanoic acid, in which there is no statistical difference between varieties. Although the pitahaya peels are a source of various compounds, the presence of fatty acids is found in a lower proportion compared to other parts of the fruit such as the seeds, where the percentage of oil varies between 18.33–28.7%, *H. polyrhizus* and *S. undatus*, respectively, observing that, like the peel, the seeds of the *S. undatus* variety have a higher percentage of fatty acids compared to *H. polyrhizus* [31]. Additionally, both parts of the fruit coincide in the identification and predominant proportion of some fatty acids such as palmitic acid and linoleate acid precursor of linoleic acid. Additionally, other investigations such as Wu et al. [26,32] showed that the fatty acids and secondary metabolites present in the peel of *H. polyrhizus* prolong the useful life of the fruit, causing a delay in senescence and fruit decomposition as well as reducing diseases in the fruit. Some of the acids found by these authors were β-linalool, palmitoleic acid, 2-hydroxy-cyclopentadecanone, and organic acids [13]. Jerônimo, et al. [33] also showed that the predominant acids in *H. undatus* seed were oleic, linoleic, and palmitic with a percentage of approximately 60%, 21%, and 13%, respectively. Likewise, the presence of high contents of linoleic, linolenic, cis-vaccenic, palmitic, and oleic acids was identified in red and white pitahaya oil extracts [34]. For this reason, the study and identification of volatile compounds are of greater importance for the food industry and to take advantage of fruit waste to take advantage of both the extraction of oils and apply various biotechnological processes to reuse the product.

### 2.3. Analysis of the Antioxidant Capability by DPPH (2,2-Diphenyl-1-picrylhydrazyl) and ABTS (2,2’-azino-bis-(3-Ethyl Benzothiazolin-6-ammonium Sulphonate) Methods in Pitahaya Peels

The antioxidant capacity of the two pitahaya varieties evaluated in this study is shown in Table 3. The *S. undatus* variety has lower antioxidant activity by the DPPH method compared to ABTS methods (23.81% and 51.22%, respectively) in the same way behaved the *H. polyrhizus* variety (24.88% and 50.92%, respectively). The data do not show a significant difference according to the analysis. However, the ABTS method shows a higher antioxidant activity compared to the DPPH method in both varieties. These data can be attributed to the nature of the compounds present in the pitahaya peels. This is attributed to the ability of ABTS to capture lipophilic and hydrophilic compounds that are involved in free radical scavenging.

One of the strategies that contributes to the reduction of free radicals is antioxidant compounds, which are present in fruits and vegetables, as well as in parts of these that are not edible or used in processes such as in the case of peels of various fruits. Studies indicate that fruit peels are a good source of antioxidants and that they are used in the food industry as an alternative to the use of synthetic antioxidants that cause health damage. The data indicate that the pitahaya *S. undatus* peel presents a lower percentage of antiradical activity by the DPPH method (23.8%) compared to the ABTS method (51%), behavior similar to that presented by the variety *H. polyrhizus* (24.8% DPPH and 50.9% ABTS), data lower than those reported by Som et al. [24] for a methanolic extract of the *S. undatus* variety for the DPPH method. Both ABTS and DPPH methods indicate the capacity of the sample to eliminate free radicals; however, the DPPH method has been related to the presence of phenolic acids and most water-soluble compounds, while the ABTS method has been related to the presence of flavonoid compounds and fatty acids, as it has the ability to absorb lipophilic compounds, which coincides with the data obtained in this study, since both the *S. undatus* and *H. polyrhizus* varieties contain a higher concentration of total flavonoids as well as the presence of fatty acids. Therefore, they showed a greater capacity to reduce the absorption of ABTS.

### 2.4. Analysis of Bioactive Compounds of Pitahaya Peels

The inhibition data of the enzymes involved in the first phase of carbohydrate degradation, as well as in the enzyme responsible for the development of hypertension, are shown in Table 4. It is observed that the peel of *H. polyrhizus* generates a greater inhibition of the alpha-amylase enzyme (67.78%) compared to *S. undatus* (57.95%); while no statistical difference is observed in the inhibition presented for alpha-glucosidase for both *H. polyrhizus* and *S. undatus*, 69.6% and 55.08%, respectively. The trend is observed in the same way in the activity of the angiotensin-converting enzyme (ACE), where the varieties do not influence the percentage of inhibition of this enzyme. The data indicate that both *H. polyrhizus* with 97.95% and *S. undatus* with 91.58% contribute with metabolites capable of inducing inhibition of enzymes involved in the development of chronic diseases.

Therefore, the inhibition of the enzymes α- amylase and α-glucosidase favor the control of chronic degenerative diseases such as diabetes mellitus. Studies carried out by Coral Caycho et al. [35] presented inhibitory changes in the pulp of yellow pitahaya and other authors such as Wang et al. [36] showed changes in inhibitors in the enzymes α-amylase and α-glucosidase compared to the most common drug for the control of diabetes mellitus. These two authors indicate that the phenolic compounds are the main compounds that intervene in this inhibition, together with the carotenoids present in some varieties of pitahya. On the other hand, the hypoglycemic effects of the fruits of the Hylocereus family can improve insulin resistance at the genetic level in fibroblasts (FGF21), regulating the metabolism of lipids and glucose. Likewise, Shad et al. [37] in the pitahaya cascade tested different pH conditions that activated the activity of the enzymes to the maximum, giving them importance in the biotechnological industry to be used as by-products in the processing of beverages and starches, etc. [37]. In addition, studies have shown that pitahaya the *H. undatus* peel inhibits lipase activities, improves the lipid profile, and helps counteract hyperlipidemia [38].

## 3. Materials and Methods

### 3.1. Sample Preparation

The pitahaya (*Hylocereus undatus* or *Selenicerus undatus* and *Hylocereus polyrhizus*) analyzed for this study were grown free of fertilizers and pesticides. The fruits were collected in a state of maturity, in Marroquin, Apaseo el Alto, Guanajuato-México, in the following coordinates: latitude 20.5167, longitude 100.5667. The fruits were washed under running water, and the pulp was separated from the peel. The peel was lyophilized for 24 h at −50 °C and 0.012 atm in a vacuum system. After the drying process, they were weighed to obtain their yield. The extraction will be carried out according to the methodology described by Cardador-Martínez et al. [39], 25 mg of dry samples were placed, and 2.5 mL of methanol was added to each sample. They were kept free of light and shaken for 24 h. Centrifuging (Thermo Scientific, Waltham, MA, USA) at 5000× *g* rpm for 10 min at 4 °C, the pellet formed at the bottom was eliminated, leaving the supernatant.

### 3.2. Total Phenolic Compounds

Folin–Ciocalteu spectrophotometric method [40] was modified for use in a 96-well microplate to determine the total phenol. The absorbance was measured at 760 nm and the results were expressed as mg equivalent of gallic acid/g of sample.

### 3.3. Total Flavonoids

The spectrophotometric method used to measure total flavonoids was determined by Oomah et al. [41]. The total flavonoids was analyzed in triplicate and the absorbance used in this method was 404 nm, and the results were expressed in routine hydrate mg equivalent/g sample.

### 3.4. Total Tannins

Total condensed tannin samples were evaluated according to the procedure described by Feregrino-Pérez et al. [42], modified for use in a 96-well microplate. The sample analysis was carried out in triplicate, using an absorbance of 492 nm, and the results were shown in equivalent mg of (+) catechin/g.

### 3.5. Identification of Gas-Chromatography-Mass Spectrometry (GC-MS) Analysis

GC-MS analysis was carried out as accomplished by Lim et al. [31] with some modifications. The supernatants of the sample were prepared with the solvent and derivatized and subsequently were taken, and 1 μL of the sample was injected in triplicate in an Agilent gas chromatograph (GC) series 7890A (Wilmington, DE) coupled to a single quadrupole mass spectrometer (MS) detector (Agilent 5975C) equipped with an electron impact (EI) ionization source. The carrier gas (helium) flow rate was maintained at 1 mL min^−1^. The injector temperature was set at 250 °C in splitless mode. An HP-88 capillary column (30 m × 0.25 mm inner diameter × 0.25 μm) was used. The initial oven temperature was 50 °C, held for 1 min, and raised to 175 °C at 15 °C min^−1^, then raised to 240 °C at 1 °C min^−1^, and held for 5 min. EI energy was set at 70 eV, and the mass range was set at m/z 50−1100. FAMEs were identified and quantified by comparison with a standard Supelco 37 Component FAME Mix, and data processing was performed using ChemStation (Agilent Technologies) software.

### 3.6. Antioxidant Capacity

#### 3.6.1. DPPH Method Antioxidant Capacity

The DPPH (2,2-Diphenyl-1-picrylhydrazyl) method was accomplished by Zenil et al. [43]. The analyses were performed in triplicate. The results will be shown in mg equivalent of trolox/g of sample, and absorbance of 520 nm.

#### 3.6.2. ABTS Method Antioxidant Capacity

ABTS (2,2’-azino-bis- (3-ethyl benzothiazolin-6-ammonium sulphonate)) was used to measure antioxidant capacity accomplished by Re et al. [44]. Analyses were measured at 734 nm, and in triplicate. The results were shown in mg equivalent of Trolox/g of a sample.

### 3.7. Biological Activity

#### 3.7.1. α-. Amylase Inhibition

The procedure accomplished by Meza and Valdés [45], used for the quantification of α-amylase, was modified in a 96-well microplate at an absorbance of 540 nm and the sample analyzed in triplicate.

#### 3.7.2. α-. Glucosidase Inhibition

Ranilla et al. [46]’s method was applied to quantify the inhibition of α-glucosidase, with an absorbance of 405 nm and performed in triplicate; this method was modified for 96-well microplates.

#### 3.7.3. Antihypertensive Activity

ACE (angiotensin-converting enzyme) inhibitions were evaluated according to the method accomplished by Salazar Aranda et al. [47], modified to a 96-well microplate, analyzing results in triplicate at an absorbance at 345 nm at 5 min^−1^.

### 3.8. Statistical Analysis

Pitahaya species were expressed as the mean of three repetitions with their standard deviation (SD). Statistical calculation was performed using the IBM SPSS Statistics 25.0 (Document number 589145; IBM Corporation 2021, Armonk, NY, USA). Significant differences at *p* < 0.05.

## 4. Conclusions

Currently, the pitahaya or dragon fruit is widely consumed due to the color of its pulp. Pulp, seeds, and peel are very rich sources of primary and secondary metabolites. The peel residues have always been considered food waste; in the pitahaya, the peel represents a third of the weight of the fruit, in addition to its high content of phytochemical compounds with a bioactive effect on chronic degenerative diseases. The nutraceutical compounds found in the pitahaya peel could be used as enzyme modulators by demonstrating in this study that they have an effect against enzymes such as α-amylase and α-glucosidase, ACE. It has been shown that pitahaya peels should not be converted into a food residue, but rather a by-product for the development of new products, because they are rich in some fatty acids, and phenolic compounds, and due to their enzymatic activity, they could be used in industry, pharmaceutical, medical, and food. In addition, in the biotechnological part, the pitahaya peel could be used for the recovery and purification of enzymes such as the amylase enzyme, since it is industrially very important in the fermentation of food, in the cosmetic, pharmaceutical, and food areas, and to convert the pitahaya peel into a sustainable bioresource. Furthermore, the results of this study showed a higher amount of flavonoid-type secondary metabolites, as well as a higher antioxidant capacity in the *S. undatus* variety, caused by the reaction that the reagent has on all the hydrophilic and lipophilic (hexadecanoic acid methyl ester and methyl linoleate) antioxidant compounds. The presence of these bioactive compounds in the pitahaya peel contributes to the inhibitory effect on enzymes related to states of hyperglycemia and hypertension that had not been described so far in the pitahaya peel. Additionally, this work is also a pioneer in the information of methylated acids that contribute to the valuation of pitahaya fruits as well as to increase the useful life of the fruit and increase the reuse of the components of by-products considered waste, now innovating in the technologies of extraction, and offering consumers products that come from recovery biomolecules that contain health benefits.

## Figures and Tables

**Figure 1 molecules-27-05339-f001:**
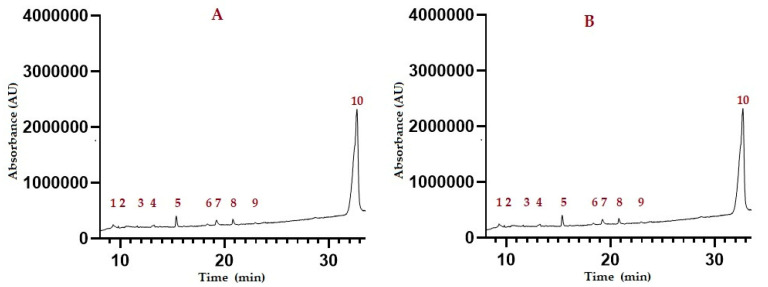
Chromatogram of pitahaya samples. (**A**) Chromatogram of pitahaya *Selenicerus undatus*. (**B**) Chromatogram of pitahaya *Hylocereus polyrhizus*.

**Figure 2 molecules-27-05339-f002:**
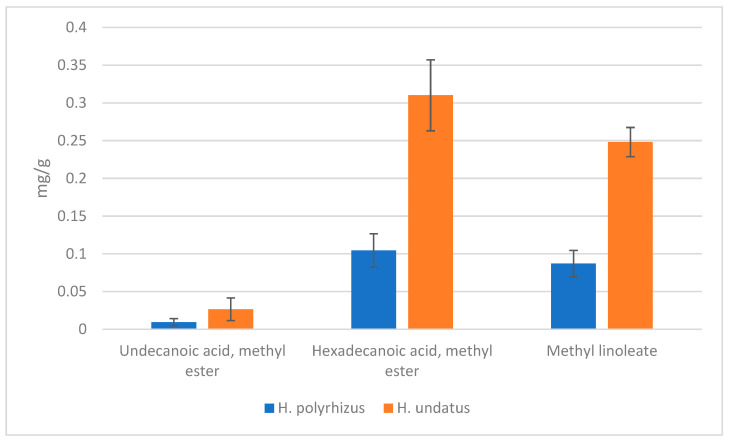
Identification of Gas-chromatography-mass spectrometry (GC-MS) analysis in *Selenicerus undatus* and *Hylocereus polyrhizus*. The average represents the value of 3 repetitions.

**Table 1 molecules-27-05339-t001:** Phenolic compounds content in methanol extracts in *Selenicerus undatus* and *Hylocereus polyrhizus.*

Peels	Phenolic mg GAE/g Sample			Flavonoids mg RE/g Sample						Tannins mg CE/g Sample						Total Phenolicmg /g Sample
		F	*p*-Value *					F	*p*-Value *					F	*p*-Value *	
** *S. undatus* **	2.717 ± 0.0319 ^a^	1.492	0.706	356.74 ± 0.0216 ^a^				0.471	0.000	2.886 ± 0.001 ^a^				4.046	0.000	362. 568 ± 0.026 ^a^
** *H. polyrhizus* **	1.863 ± 0.0053 ^b^			352.09 ± 0.0455 ^a^						1.628 ± 0.002 ^b^						355..033 ± 0.038 ^b^

mg GAE/g sample (mg Gallic acid equivalents/ g sample), mg RE/g sample (rutine equivalents/g sample), mg CE/g sample (mg catechin equivalents/g sample). The average represents the value of 3 repetitions. * Comparison between means (T Student α ≤ 0.05). Means with different letters in the same column are statistically different.

**Table 2 molecules-27-05339-t002:** Retention times and areas measured by the team, FAMEs in two varieties of pitahaya *Selenicerus undatus* and *Hylocereus polyrhizus*. T_R_(retention time). In the spaces where a hyphen was placed is an unidentified element.

Peak	TR		Compound Name		Area%		% Similarity to Library	
	*S.* *undatus*	*H. polyrhizus*	*S.* *undatus*	*H. polyrhizus*	*S.* *undatus*	*H. polyrhizus*	*S.* *undatus*	*H. polyrhizus*
1	9.27	8.750	9-Octadecenamide	Heneicosane	1.30	5.85	43	87
2	9.78	9.26	Ethanedioic acid, dimethyl ester	Nonadecane	0.58	6.56	80	91
3	11.615	11.615	Undecanoic acid, methyl ester	Undecanoic acid, methyl ester	5.51	5.51	92	92
4	13.94	13.25	-	Methyl tretadecanoate	-	1.28	-	95
5	15.35	15.388	Hexadecanoic acid, methyl ester	Hexadecanoic acid, methyl ester	51.29	10.38	98	97
6	16.04	16.04	3-Acetoxy-3-hydroxypropionic acid, methyl ester	Dimethyl dl-malate	0.40	0.41	72	83
7	19.88	18.98	3-Hexadecanol	Eicosane	0.70	15.01	35	87
8	20.78	20.82	Methyl linoleate	Methyl linoleate	32.08	6.20	99	99
9	23.84	22.94	Decanamide	9,12,15-Octadecatrienoic acid, methyl ester	0.65	2.15	53	98
10	34.35	34.78	9-Octadecenamide	Hexadecanamide	62.35	5.09	96	81

**Table 3 molecules-27-05339-t003:** Antioxidant capability in methanol extracts of peels in *Selenicerus undatus* and *Hylocereus polyrhizus*.

Peels		DPPH%					ABTS%					
					F	*p*-Value*					F	*p*-Value*
*S. undatus*	23.81 ± 0.46 ^a^				23.84	0.000	51.22 ± 0.171 ^a^				71.79	0.000
*H. polyrhizus*	24.88 ± 0.29 ^a^						50.92 ± 0.421 ^a^					

The average represents the value of 3 repetitions. * Comparison between means (T Student α ≤ 0.05). Means with equal letters in the same column are not statistically different.

**Table 4 molecules-27-05339-t004:** Biological activity of secondary metabolites in methanol extracts in melissa, peppermint, thyme, and mint.

Peels	α-Amylase%						α-Glucosidase%						ACE%					
					F	*p*-Value *					F	*p*-Value *					F	*p*-Value *
*S. undatus*	57.95 ± 6.39 ^a^				10.137	0.000	55.08 ± 1.39 ^a^				4.596	0.000	91.5895 ± 0.65 ^a^				33.107	0.000
*H. polyrhizus*	67.78 ± 6.06 ^b^						69.60 ± 1.02 ^b^						97.9571 ± 7.44 ^a^					

The average represents the value of 3 repetitions. * Comparison between means (T Student α ≤ 0.05). Means with equal letters in the same column are not statistically different.

## Data Availability

Not applicable.

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
