# Peer review of "Pitahaya Peel: A By-Product with Great Phytochemical Potential, Biological Activity, and Functional Application"

_molecules, 2022, doi:10.3390/molecules27165339_

Round 1

Reviewer 1 Report

I consider that the article is now ready for publication

Author Response

There are no comments from the reviewer, thanks for your very accurate consideration of the article

Reviewer 2 Report

Research work resubmitted by Jimenez-Garcia et al entitled “Pitaya peel: a by-product with great phytochemical potential, biological activity, and functional application” is a preliminary study. After corrections made by authors, manuscript has been improved but still it is not suitable for publication in present form. Find mentioned below some of the suggestions:

1.      The abstract still needs improvement. Methodology and results should be discussed in abstract section with some data.

2.      In line 122, western India should be corrected.

3.      Check the sentence in line 225.

4.      Check the sentence in line 230.

5.      Check the sentence in line 259-266.

6.      Check the use of present, past and future tenses.

7.      In line 423-424, check the values of phenolic compounds, 2,717 and 1,863 mg GAE/g sample or 2.717 and 1.863.

8.      In line 425, no need to mention all authors names.

9.      Check the sentence in line 831-832.

10.  Mention complete list of compounds in table form using the relative retention index.

11.  GC chromatogram should be mentioned.

12.  In line 764, why GC-MS mentioned in brackets?

13.  Still English has to be corrected by some professionals or native English speakers. In line 764-765, the sentence should be corrected as “GC-MS analysis was carried out as accomplished by Lim, et al. [31] with some modifications”.

14.  Check the use of abbreviations throughout the manuscript, for example in line 562, DPPH and ABTS should be written in full form.

Author Response

We tried to attend to the observations since the outline of the article and the outline indicated by the reviewer do not coincide in the attached file, there are fewer outlines than the indications given by the reviewer. That is why we tried to attend to it by verifying the idea of the paragraph. A revision of the English was also carried out, modifying some verbs, modifying time and also modifying some sentences, and homologating some words.

Q1.- The abstract still needs improvement. Methodology and results should be discussed in the abstract section with some data.

Answer.- The abstract has been modified, as far as possible since it must comply with 200 words, for this reason only the values in the results and a little more discussion were added.

Q2.- In line 122, western India should be corrected.

Answer.- The lines mentioned by the reviewer do not coincide with those found in the attached article. I try to correct looking for I am discriminated against by the authors. In line 122, "india" is changed to "India". and the error was identified in the gender in the sentence

Q3.- Check the sentence in line 225.

Answer.- The line was identified and a correction was made to the line that was considered to correspond, which is in our file on lines 89-92.

Q4.- Check the sentence in line 230.

Answer.- The line was revised and the change in the sentence was made in the corresponding line

Q5.- Check the sentence in lines 259-266.

Answer.- The line was revised and the change in the sentence was made in the corresponding line

Q6.- Check the use of present, past, and future tenses.

Answer.- The entire article was revised in the English language, adjusting the tenses of the verbs throughout the manuscript. The line was modified 35, 46, 78, 88, 171, 181, 248.

Q7.- In lines 423-424, check the values of phenolic compounds, 2,717 and 1,863 mg GAE/g sample or 2.717 and 1.863.

Answer.- The values of the phenolic compounds were revised and the comma was changed to a point to standardize the wording.

Q8.- In line 425, no need to mention all authors' names.

Answer.- The indicated correction was made, only that the reference editor attached it

Q9.- Check the sentence in lines 831-832.

Answer.- The line was revised and the change in the sentence was made in the corresponding line

Q10.- Mention complete list of compounds in table form using the relative retention index.

Answer.- The table of the compounds identified by CG-MS with retention time and area % is attached.

Q11.- GC chromatogram should be mentioned.

Answer.- The chromatogram of the samples is attached.

Q12.- In line 764, why GC-MS mentioned in brackets?

Answer.- Thank you for your observation, the parentheses are removed.

Q13.- Still English has to be corrected by some professionals or native English speakers. In line 764-765, the sentence should be corrected as “GC-MS analysis was carried out as accomplished by Lim, et al. [31] with some modifications”.

Answer.- The English in the article was corrected by some professionals and the identification method was modified by GC-MS trying to summarize this so as not to only place the author based on the methodology.

Q 14.- Check the use of abbreviations throughout the manuscript, for example in line 562, DPPH and ABTS should be written in full form.

Answer.- The definition of the abbreviation was placed in the first position of appearance in the article.

Reviewer 3 Report

In the resubmitted manuscript the authors have addressed most of the reviewer comments both in terms of the manuscript form and the experimental discussion. The title has been changed, the abstract and the introduction have been rewritten to provide a more solid justification to the work. The present version of the manuscript is improved, and I have no further points that require clarification. Therefore, I do recommend publication in Molecules.

Author Response

There are no comments from the reviewer, thanks for your very accurate consideration of the article

This manuscript is a resubmission of an earlier submission. The following is a list of the peer review reports and author responses from that submission.

Round 1

Reviewer 1 Report

In the present manuscript the authors report the extraction and quantification of the polyphenolic compounds, biological activities of interest, and low molecular weight compounds present in the skin of the two varieties of pitahaya. The manuscript flows smooth, and the introduction provides a clear picture and justification of the work. The references are appropriate to a full paper and the discussion is solid. Nevertheless, few points, listed below, need to be addressed before recommending publication in Molecules.

Major points:

-          The data analysis should be described more in details rather than just describing the values reported in the tables. Trying to correlate the obtained concentration with the degree of maturation, locations, or differences between the two varieties for the results described in section 2.1 and 2.2.

-          Are the obtained results comparable with data already available?

-          The material and methods section should be revised to include more experimental details. The authors reported references of each applied protocol, but at the same time, claim to have modified the protocol for a 96 well-plate. Since modification have been applied, detailed protocols should be described to provide reproducibility of the results.

-          Page 7 line 151 “The S. undatus variety has a higher antioxidant activity by both the DPPH and ABTS” this statement is in contrast with the reported number since the DPPH method for the H. polyrhizus showed a higher value.

-          Why the ABTS methods shows higher activity in comparison to DPPH?

Minor points:

-          Page 3 line 87 “for interest” should be deleted

-          Page 4 line 100 “A few years to date, Studies” should be revised as “A few years to date, studies”

-          Page 14 line 255 “palmitic acid >Methyl linoleate>Undecanoic acid” should be revised as “palmitic acid > methyl linoleate> undecanoic acid”

-          Page 15 line 296 “The extraction will be carried out” should be revised as “The extraction was carried out”

-          Page 16 line 299 “5000rpm” should be revised as “5000 rpm”

-          Page 16 line 303 “with modified for use” should be revised as “modified for use”

-          Page 16 line 310 “was 4004 nm” should be revised as “was 404 nm” or “was 400 nm”

-          Page 17 line 327 “mLmin−1” should be revised as “mL min−1

-          Page 17 line 329 “min−1” should be revised as “min−1

Reviewer 2 Report

Research work submitted by Jimenez-Garcia et al entitled “Biological Activities in Pitahaya Peels (Hylocereus spp)” is a preliminary study. In my opinion, the manuscript is not suitable for publication in its present form. Find mentioned below some of the suggestions:

1.      Title of the manuscript should be changed, it should reflect the study.

2.      The abstract should be rewritten; methodology and results should be discussed in the abstract section with some data.

3.      In the abstract, line 29, H. polyrhizus and S. undatus should be written in full form.

4.      In the abstract, line 38, Abbreviation ABTS should be mentioned in full form.

5.      Extensive English corrections needs to be done.

6.      What is the basis of analysis of fatty acid profile only? What about other constituents? Mention a complete list of compounds in table form using the relative retention index.

7.      GC chromatogram should be mentioned.

8.      In section Total flavonoids methodology, line 310, the absorbance used was 4004nm?

9.      Line 321, GC-MS, why in brackets?

10.  Line 321, reference 29, should be Lim et al., instead of writing all author names.

Reviewer 3 Report

The article entitled "Biological Activities in Pitahaya Peels (Hylocereus spp)" in my opinion is not yet ready for publication in this journal.

There are many problems, but I especially highlight the statistical analysis. It was not clear how she was executed. The authors report that they did ANOVA and Tukey's Test.

How is it possible to perform these tests with only 2 menas to compare? The T test is indicated for comparison between 2 means, for more than 2, we do the ANOVA and after the Tukey test. Authors should clarify this further in the text.

Overall, the paper only compares 2 species of Pitahaya peels with respect to various parameters, but does not have a clear application of this result. What is the innovation or novelty of the paper? This should be clearer in writing.

Reviewer 4 Report

 In this study, the total phenolic compounds, total flavonoids, total tannins and fatty acids were determined and the Antioxidant capacity and biological activity were evaluated in Pitahaya Peels. This work provide base for the use of Pitahaya Peels. This work is important and interesting. However, some revision is required and the following points should be addressed

1.       The abstract is long and not good and should be rewritten

2.       The introduction of the manuscript should be improved.

3.       The GC-MS spectrum should be provided

4.       The logic of the article is confusing. I suggest that the results section and discussion section should be combined as results and discussion.

5.       The contribution of a chemical substance to biological activity is not clear. Could you please clarify the relationship between them.

6.       The English of the manuscript should be improved.